# Development and Characterization of a Biodegradable PLA Food Packaging Hold Monoterpene–Cyclodextrin Complexes against *Alternaria alternata*

**DOI:** 10.3390/polym11101720

**Published:** 2019-10-21

**Authors:** Velázquez-Contreras Friné, Acevedo-Parra Hector, Nuño-Donlucas Sergio Manuel, Núñez-Delicado Estrella, Gabaldón José Antonio

**Affiliations:** 1Campus de los Jerónimos, Universidad Católica San Antonio de Murcia, 135, 30107 Guadalupe, Murcia, Spain; fvelazqu@up.edu.mx (V.-C.F.); enunez@ucam.edu (N.-D.E.); 2Universidad Panamericana ESDAI, Álvaro del Portillo 49, Zapopan 45010, Jalisco, Mexico; hector.acevedo@up.edu.mx; 3Departamento de Ingeniería Química, Universidad de Guadalajara, Blvd, Marcelino García Barragán 1421, Guadalajara 44430, Jalisco, Mexico; gigio@cencar.udg.mx

**Keywords:** food packaging, poly(lactic acid), thymol, carvacrol, β-cyclodextrin, antifungal activity

## Abstract

The fungi of the genus *Alternaria* are among the main pathogens causing post-harvest diseases and significant economic losses. The consumption of *Alternaria* contaminated foods may be a major risk to human health, as many *Alternaria* species produce several toxic mycotoxins and secondary metabolites. To protect consumer health and extend the shelf life of food products, the development of new ways of packaging is of outmost importance. The aim of this work was to investigate the antifungal capacity of a biodegradable poly(lactic acid) (PLA) package filled with thymol or carvacrol complexed in β-cyclodextrins (β-CDs) by the solubility method. Once solid complexes were obtained by spray drying, varying proportions (0.0%, 1.5%, 2.5%, and 5.0 wt%) of β-CD–thymol or β-CD–carvacrol were mixed with PLA for packaging development by injection process. The formation of stable complexes between β-CDs and carvacrol or thymol molecules was assessed by Fourier-transform infrared spectroscopy (FTIR). Mechanical, structural, and thermal characterization of the developed packaging was also carried out. The polymer surface showed a decrease in the number of cuts and folds as the amount of encapsulation increased, thereby reducing the stiffness of the packaging. In addition, thermogravimetric analysis (TGA) revealed a slight decrease in the temperature of degradation of PLA package as the concentration of the complexes increased, with β-CD–carvacrol or β-CDs–thymol complexes acting as plasticisers that lowered the intermolecular forces of the polymer chains, thereby improving the breaking point. Packages containing 2.5% and 5% β-CD–carvacrol, or 5% β-CD–thymol showed *Alternaria alternata* inhibition after 10 days of incubation revealing their potential uses in agrofood industry.

## 1. Introduction

Nowadays, the increasing consumer demand for healthy, freshly prepared, and convenient fruits and vegetables has driven the rapid growth of the fresh-cut produce industry worldwide, with benefits in multi-billion dollars [1]. In addition, new lifestyle drivers such as health and aging of population has stimulated the agrofood industry to enhance the offer and delivery of value-added products, such as minimally processed fruits and vegetables packaged in sealed polymeric films or on trays, ready for immediate consumption or direct cooking.

However, this trend has disturbed the scenario of foodborne diseases worldwide caused by pathogenic microorganisms, with important economic and social impacts [2], since fresh and minimally processed foods may undergo negative qualitative changes related to high respiratory rate, moisture loss, rapid enzymatic browning, and microbial contamination which lead to the rapid deterioration of the products [3]. In addition, fungal contamination of crops through latent infections usually occurs in the fields; nevertheless, the rotting arises later, during the storage and transport before marketing. The fungi of the genus *Alternaria* are among the main pathogens causing post-harvest diseases and significant economic losses. These fungi also represent a serious toxicological risk as they produce a broad spectrum of mycotoxins and secondary metabolites, which can cause problems in humans and animals [4].

This issue has raised considerable challenges for food packaging companies and researchers that specifically use biodegradable materials or bio-based packaging for food preservation. Despite the good properties of petroleum-based plastics, their widespread use and accumulation cause serious environmental problems and dependence on fossil resources. In fact, packaging applications contribute to 63% of the current plastic waste, and it is estimated that less than 14% are recyclable [5].

To overcome the described drawbacks, different approaches have been carried out to obtain bioplastics with analogous functionalities to petrochemical polymers. Poly(lactic acid) (PLA), a biodegradable aliphatic polyester which can be obtained by fermentation of renewable resources such as corn, tapioca, and sugarcane [6], meets several requirements such as high mechanical strength, biodegradability, biocompatibility, bio-absorbability, transparency, low toxicity, and easy process ability [7] to be thoroughly employed in agricultural films, biomedical devices, and food packaging [8,9] and used as a suitable carrier of active compounds to yield antioxidant or antimicrobial effects [10,11].

Currently, consumer concerns about the potential toxicity to humans of synthetic antimicrobials such as butylated hydroxytoluene (BHT) or butylated hydroxyanisole (BHA) have resulted in the increased use of natural antimicrobials, which receive a good deal of attention for a number of microorganism control issues [12]. As a result, different antimicrobials have been added to different packaging materials. In particular, essential oils and their bioactive molecules such as carvacrol and thymol have been thoroughly tested in vitro [13,14] or in different food systems such as meat, dairy or vegetable samples [15] due to their insecticidal, antiviral, antimicrobial, and antifungal activities [16]; however, their high volatility and reactivity limits their application as food preservatives. In fact, long storage time and temperature could magnify volatilization and drastically lessen their activity, requiring as consequence high concentrations to ensure antimicrobial activity, which is a detrimental praxis for organoleptic attributes (flavor, taste, and aroma) and acceptability of foods, so this strategy is not considered in practice.

In order to increase the applicability of natural antimicrobial formulations, these drawbacks could be overcome by microencapsulation or complexation techniques using cyclodextrins (CDS), which are cyclic oligosaccharides derived from starch made up of 6, 7, or 8 units of D-glucose monomers linked by α(1,4) bonds, shaped as a truncated hollow cone [17] that presents an internal hydrophobic cavity to interact with non-polar active constituents of essential oils or their bioactive molecules such as carvacrol and thymol, whereas the external face is hydrophilic, improving their water solubility and gradually increasing their effectiveness using lower concentrations of these compounds.

As a preliminary stage to subsequently evaluate the antifungal capacity of a biodegradable poly(lactic acid) (PLA) package carrying as preservatives carvacrol or thymol complexed in CDs (as described here), their complexation was carried out with native and modified CDs [18] and the antimicrobial and antifungal effects of their respective complexes was verified by comparison with hydroxypropyl-β-cyclodextrins (selected due to their highest Kc values) against *Escherichia coli*, *Staphylococcus aureus*, and *Galactomyces citri-aurantii* [19,20]. However, only native CDs (α, β, and γ) are considered as GRAS (generally recognized as safe) and are the only ones authorized to come into contact with foods.

Therefore, the present study focuses on the design and optimization of a controlled release system of antifungal carvacrol or thymol volatiles encapsulated in β-CD to be incorporated into a biodegradable polymeric matrix of PLA by industrial injection. The optimization of stable complexes between β-CDs and carvacrol or thymol molecules and characterization by Fourier-transform infrared spectroscopy (FTIR) were carried out. Mechanical, structural, and thermal characterization of developed packaging was carried out and materials behavior against *Alternaria alternata* growth was also investigated.

## 2. Materials and Methods

### 2.1. Materials

Carvacrol (CAS: 499-75-2, 99.5% purity), thymol (CAS: 89-83-8, 98.7% purity), and β-cyclodextrin (β-CD >95%, food grade) were purchased from Sigma-Aldrich Corp (Saint Louis, MO, USA). The chemical structures of the two monoterpenes are shown in Figure 1. Poly(lactic acid) (PLA, Ingeo™ Biopolymer ref. code: 3251D) with a weight-average molecular weight ($\bar{M}$w) of 5.5 × 10^4^ g/mol, polydispersity index (PI) of 1.62, and low D-isomer content (99% l-lactide/1% D-lactide), provided by PromaPlast Co (Guadalajara, Jalisco, Mexico) and manufactured by Nature Works LLC (Blair, NE, USA), was selected for injection moulding applications since it has a higher flow capability (relative viscosity 2.5, glass transition temperature Tg = 55–60 °C, melting temperature Tm = 155–170 °C, and processing temperature 188–210 °C) than other resins currently available in the marketplace. *Alternaria alternata* strain ATCC 42761 (isolated from blackberries in Georgia, USA) was purchased from SENNA laboratories, Mexico City. Potato dextrose agar (PDA) was provided by Bioxon, Mexico. The rest of the chemical products were of analytical grade.

### 2.2. Preparation of β-CD Inclusion Complexes

Both β-CD–carvacrol and β-CD–thymol inclusion complexes were prepared using the solubility method [18]. For that, aqueous solutions of increasing concentrations of β-cyclodextrin (0–15 mM) were prepared in sodium phosphate buffer (100 mM, pH of 7.0) in a total volume of 100 mL. A saturating amount of carvacrol or thymol was added to each one of the solutions and kept in an ultrasound bath (Ultrasons HP, Selecta, Spain) for 60 min in the dark at 25 °C, until equilibrium was reached. After that, to remove excess monoterpene, the respective solutions were filtered through a nylon filter of 0.45 μm. Liquid complexes were used for phase solubility diagrams, determining the concentration of entrapped monoterpene by GC/MS, and posterior spray drying process to obtain powdered dehydrated complexes.

From the phase diagrams of carvacrol or thymol, complexed with β-CDs, the parameters efficiency of complexation (CE) and the molar ratio (MR) were determined. CE is the ratio between the dissolved complex and free cyclodextrin (CD) concentration. It is independent of S_0_ (aqueous solubility), and was calculated from the slope of the phase solubility profiles by using Equation (1).
(1)$$CE\;\left(\%\right)=\frac{\left[dissolved-complex\right]}{{\left[CD\right]}_{f}}$$

The MR of β-CD–carvacrol and β-CD–thymol inclusion complexes was calculated using CE values with Equation (2).
(2)$$MR\;=\;\frac{1}{\left(1+\;\frac{1}{CE}\right)}$$

### 2.3. Atomization Process to Obtain Complexes in Solid State

To obtain complexes in solid state, the β-CD–carvacrol and β-CD–thymol solutions were subjected to an atomization process using a laboratory-scale atomizer and Büchi B290 Mini Spray Dryer (Flawil, Switzerland) working with air as the carrier gas at a flow rate 5 mL/min, pressure of 3.2 bar, and an inlet and outlet temperature of 170 ± 2 °C and 68 ± 2 °C, respectively, using a 1.5 mm nozzle diameter. In each case, the entrapment efficiency (EE) was determined with respect to the theoretical number of monoterpenes present in the inclusion complex after atomization, using Equation (3).

(3)$$\mathrm{EE}\;=\;\frac{\mathrm{Amount}\;\mathrm{of}\;\mathrm{active}\;\mathrm{compound}\;\mathrm{entrapped}\;}{\left(\mathrm{Initial}\;\mathrm{active}\;\mathrm{compound}\;\mathrm{amount}\right)}\;\times \;100$$

Furthermore, the process performance (PP) was determined as follows:(4)$$\mathrm{PP}\;=\;\frac{\mathrm{Total}\;\mathrm{weight}\;\mathrm{obtained}\;\mathrm{from}\;\mathrm{solids}\;\mathrm{after}\;\mathrm{spray}\;\mathrm{drying}\;\mathrm{process}\;\left(g\right)\;}{\left(\mathrm{Initial}\;\mathrm{Initial}\;\mathrm{weight}\;-\mathrm{CD}\;\mathrm{in}\;\mathrm{solution}\;\left(g\right)\;\right)}\;\times \;100$$

Carvacrol and thymol concentrations in dehydrated complexes were quantified after spray drying. For that, β-CD–carvacrol and β-CD–thymol were diluted in ethanol (complex: ethanol, 20:80, *v*/*v*), to break the complexes formed. After that, β-CDs was removed from the solution, leaving only the active compound for further quantification in triplicate, by GC/MS analysis at Agilent Technologies 7890B (Palo Alto, CA, USA) coupled to a 5977A mass spectrometer, as previously described by Rodríguez-López et al. (2019) [18].

### 2.4. Fourier-Transform Infrared Spectroscopy (FTIR)

The FTIR spectra used to study changes of chemical structures of free carvacrol and thymol, and their respective complexes were acquired using a Varian FTIR 670 (Agilent Tech, Amstelveen, The Netherlands) spectrophotometer coupled with an accessory to analyze the attenuated total reflectance (ATR) with a wave number resolution of 0.10 cm^−1^ in the range of 250–4000 cm^−1^. A minimum of 32 scans were signal-averaged with a resolution of 4 cm^−1^ in the above ranges.

### 2.5. Boxes Production

The PLA samples were dried in an oven at 60 °C for 4 h to avoid bubbles in the molding process. After that, physical mixtures were performed using as ingredients PLA (100%, 98.5%, 97.5%, and 95% weight percentages, wt%), and dehydrated complexes of β-CD–carvacrol or β-CD–thymol at (0%, 1.5%, 2.5%, and 5% wt%), that were introduced in a pilot extruder to produce pellets. The extruder had a screw diameter (D) of 25.4 mm, screw length (L) of 406.4 mm (L/D ratio of 16), four heating zones, and a slot 1.75 mm matrix outlet. The barrel temperature profile was set at 150/170/180/180 °C with a screw speed of 30 rpm.

The pellets produced in the previous step were thermo-pressed in the Belken BLD-68 injector from AG Plastic (Querétaro, México), optimizing the parameters of heated mold (180 °C/100 bar), to ensure the adequate fluidity of the material to produce (12 × 10 × 3.0 cm) boxes (Figure 2). Once the material reached the cooling temperature, the boxes were then released from the molds.

### 2.6. Packaging Characterization

#### 2.6.1. Mechanical Properties

Tensile proofs were carried out in the universal traction machine SFM 100 from United Testing Systems (Ontario, Canada). Previously the packages were manually cut to obtain assay pieces (ten of each formulation), according to dimensions established by the ASTM method D-638. The tests were conducted at room temperature, at 5 mm/min speed using an initial grip length of 25 mm. The parameters, namely, average of maximum stress (MPa), breaking point (%), and Young’s modulus (MPa) were determined for the pieces of PLA and β-CD–carvacrol or β-CD–thymol, according to the aforementioned procedure [21].

#### 2.6.2. Scanning Electron Microscopy (SEM)

The structure of the packaging material was determined by scanning electron microscopy MIRA3 model form TESCAN (Brno, Czech Republic). Packages were previously frozen at −80 °C, manually fractured, and later placed on the slide and gold coated during 90 s using a sputter coater. All the samples were evaluated using a voltage of 7.0 kV.

#### 2.6.3. Thermal Characterization of the Developed Packaging

The thermal evaluation of the packaging material was done by differential scanning calorimetry (DSC) and thermogravimetric analysis (TGA). DSC assays were performed in DSC-Q100 (TA Instruments, New Castle, USA). Firstly, pieces of 5 mg were dried for 48 h in an oven at room temperature; after that, samples were placed in an aluminium capsule that was subjected to a temperature scan from 20–230 °C at a heating rate of 10 °C/min under inert nitrogen atmosphere. In addition, thermal stability of the materials was carried out by TGA using the gravimetric thermal analyzer TGA-550 (TA Instruments, New Castle, PA, USA). For that, samples of 10 mg were weighed and placed in platinum trays, which were subjected to a temperature scan of 20–600 °C at a heating rate of 20 °C/min under a nitrogen atmosphere.

### 2.7. Antimicrobial Activity

The antifungal activity of the packages with β-CD–thymol and β-CD–carvacrol was evaluated by vapor phase diffusion, in triplicates, according to Du et al. [22] using a strain of *A. alternata* (ATCC 42761). Pure fungal cultures in potato-dextrose agar medium plates with 14 days of incubation (23 °C) were suspended in 10 mL sterile distilled water containing 0.05% of Tween 20, and collected by gently scraping the surface of the agar with a sterile L-shaped glass rod. Next, the arthrospores concentration was adjusted to 10^6^ spores/mL using the McFarland scale (Shumadzu, UVmini-1240), and the inoculum was used for in vitro bioassays.

For the bioassay, 3.0 µL of spore suspensions were placed in the centre of Petri dishes previously filled with inoculated potato dextrose agar (PDA). Subsequently, these boxes were incubated at 25 °C for 5 and 10 days, inverted and covered with parafilm, and were used as controls.

Packages containing different concentration of active compounds were aseptically cut into 50 mm rectangles and placed on top of the Petri dishes. Parafilm M (Bemis) was used to hermetically seal the Petri dishes, which were incubated at 25 ± 1 °C in an incubator (Binder ED), for 120–240 h. After the incubation period, the inhibition zone diameter created by the vapor and active compound (thymol or carvacrol complexed with CDs) released from the packaging into the culture medium was measured and related to the package antimicrobial activity.

The growth of fungal cultures as well as controls were daily evaluated by measuring the diameter of the colony or surface area (diameter at right angles to each other) of the plates occupied by the colony during incubation time. The measurements were carried out with a gauge on the agar surface reporting growth at 5 and 10 days. Due to the transparency of the materials used, these measurements were conducted without opening the box. Every assay was tested in triplicate and the results were statistically analyzed.

### 2.8. Statistical Analysis

The data corresponding to mechanical properties and the diameter of the colony in the antifungal activity were subjected to statistical analysis. Analysis of variance (ANOVA) and Tukey’s multiple comparison test were performed using MINITAB 18 statistical software (Paris, France), at a 5% significance level.

## 3. Results and Discussion

### 3.1. Assessment of the Obtained Complexes

As described previously by Rodríguez-López et al., 2019 [18], phase solubility diagrams of carvacrol and thymol with β-CDs were carried out at pH 7.0 (25 °C), since the pH of the medium could condition its dissociation degree and consequently its solubility, thus determining the stability of the complexes. By using linear regression analysis of the phase solubility diagrams and considering the formation of β-CD–carvacrol and β-CD–thymol 1:1 complexes when the concentration of β-cyclodextrin was 11 mM, it was possible to determine the complexation constant (Kc), the complexation efficiency (CE), and molar ratio (MR) by applying Equations (1) and (2).

As can be seen in Table 1, β-CD–thymol and β-CD–carvacrol complexes show the same molar ratio (1:2), indicating that almost one of every two β-CDs molecules in solution is forming soluble complexes with carvacrol or thymol [18]. However, the efficiency of complexation obtained for carvacrol (105.6) is significantly higher than that obtained for thymol (69.3).

For further packaging formulations with PLA, soluble complexes of β-CD–carvacrol and β-CDs–thymol were subjected to a spray drying process to obtain complexes in a solid state to improve their management.

After the dehydration process, entrapment efficiency (Equation (3)) and process performance (Equation (4)) parameters were determined (see Table 2), showing similar values for both the parameters, but slightly higher for thymol.

FTIR is a suitable technique for evidencing the formation of the β-CD–carvacrol and β-CD–thymol inclusion complexes (Figure 3), due the shift or vanishing of the stretching and bending vibrations of the functional groups of guest molecule once complexed.

As can be seen in Figure 3a, the IR spectrum of thymol (structural isomer of carvacrol), shows several characteristics peaks: 3164 cm^−1^, O-H stretching and bending vibrations (3164 cm^−1^ and 1453 cm^−1^, respectively); C-H symmetric and asymmetric stretching bands at 2858 cm^−1^ and 2897 cm^−1^, respectively; and three C=C stretching vibrations of weak intensity at 1624 cm^−1^, 1592 cm^−1^, and 1506 cm^−1^, revealing the tri-substitution of the aromatic ring. With respect to the substituents of the aromatic ring, methyl (-CH_3_) appears at 1344 cm^−1^ and a typical doubled signal (like a tooth) at 1410 cm^−1^ characteristic of isopropyl group was observed. The IR spectrum of β-CDs (Figure 3a) showed characteristic bands corresponding to stretching vibrations of O-H and C-H links, around 3268 cm^−1^ and 2875 cm^−1^, respectively and O-H bending vibrations at 1623 cm^−1^.

In addition, Figure 3b shows an approach of the IR spectrum of β-CDs revealing C–O–C symmetric and asymmetric vibrations at 890 cm^−1^, 1170 cm^−1^, and 1021 cm^−1^; respectively. With respect to free *β*-cyclodextrin, the spectra of β-CD–carvacrol and β-CD–thymol inclusion complexes (Figure 3b) highlighted the presence of characteristics C=C peaks corresponding to carvacrol and thymol aromatic ring close to 1590 cm^−1^ and vibrations of their respective methyl (-CH_3_) groups appear at 1430 cm^−1^ (asymmetric) and 1360 cm^−1^ (symmetric). These shifts relative to those of respective free compounds provide a clear evidence of host-guest interactions.

### 3.2. Mechanical Properties of PLA Packaging Loaded with β-CD–Carvacrol or β-CD–Thymol Inclusion Complexes

In order to prevent breakages during the packaging process, polymeric materials to be used in food packaging require sufficient flexibility [23]. In this sense, the mechanical properties of the PLA boxes with different concentrations (1.5%; 2.5%, and 5%, wt%) of β-CD–carvacrol (Table 3) and β-CDs–thymol (Table 4) inclusion complexes were evaluated, using PLA boxes without solid complexes as control.

As can be seen in Table 3, Young’s modulus ranged from 2873 to 1960 MPa for β-CD–carvacrol complexes and from 2873 to 2394 MPa for β-CD–thymol complexes (see Table 4), showing lower values than the control trays (only PLA). In fact, Young’s modulus gradually decreases as the concentration (weight percentage, w%) of the dehydrated complexes increases, obtaining the lowest value of Young’s modulus in the sample fortified with 5% of carvacrol (see Table 3), with a significant difference respect to the average value (*p* < 0.05). The same trend was observed when evaluating the maximum stress, with the lowest value being observed in the PLA package enriched with dehydrated complexes of β-CD–carvacrol (5%), 14% lower than the value obtained for PLA fortified with thymol complexes at the same concentration (w%). The different mechanical values observed when both the complexes were added to the PLA polymer could be due to the higher CE value obtained for carvacrol–β-CDs (105.6%), 65% higher than the value obtained for thymol–β-CDs (69.3%), revealing that CE values above 100% indicate that at pH 7.0, there are more β-CDs complexing carvacrol than free in solution. In the case of thymol, the number of β-CDs complexing thymol is lower, since CE is less than 100%, and in consequence, the decrease in mechanical properties is less pronounced.

Regarding the breaking point, a significant increment of this parameter was observed as the concentration (wt%) of the dehydrated complexes increased, improving 25% and 23% of the elongation capacity of the polymeric material (control), when 5% of β-CD–carvacrol or 5% of β-CD–thymol, respectively, were added to PLA. This behavior may be attributable to a plasticizing effect triggered by the addition of β-CD complexes to the polymer matrix disrupting the crystalline structure of PLA and increasing its ductile properties [24].

As a result, the relative high elongations achieved were beneficial since the boxes presented better flexibility. These results are consistent with those obtained by Ramos and López-Rubio, wherein an increase in elongation and breaking point in plastic films composed of polypropylene/carvacrol/thymol were evidenced [24,25].

### 3.3. Scanning Electron Microscopy

The fracture micrographs of the samples are shown in Figure 4.

As can be seen (Figure 4a), while control sample (only PLA) had an irregular surface, the PLA samples enriched with β-CD–carvacrol (Figure 4e–g) or β-CD–thymol (Figure 4b–d) complexes exhibited a more uniform surface as the concentration (wt%) increased. In this sense, the decrease in the number of cuts and folds of polymeric material was directly proportional to the concentration of added complex.

These results are in agreement with the values reported in mechanical tests (see Table 3 and Table 4), evidencing that the increase of the concentration of complexes in the formulation of the plastic material favors obtaining more flexible packaging (decrease in Young’s modulus), providing the formation of a smoother and continuous surface.

This fact could be due to encapsulation which helps incorporate the active compound (carvacrol or thymol) into the polymeric matrix, since different results have been described in the literature when raw essential oils (without encapsulation) were added to polymeric materials to produce heterogeneous structures with oil droplets trapped into the polymer [26,27].

### 3.4. Differential Scanning Calorimetry

To investigate the thermal transitions of the films studied, DSC measurements were accomplished. As can be seen in Table 5, the packages containing β-CD–carvacrol or β-CD–thymol complexes showed similar thermal properties, regardless of their concentration. The glass transition temperature (Tg) of the PLA-enriched materials was analogous to that obtained for PLA control, and similar to Tg values described in the literature [28], indicating that the amorphous phase of the PLA does not undergo any change.

On the other hand, the packaging with additives shows a significant variation in the cold crystallization temperature with respect to the control packaging (Tc = 102.7 °C; PLA 0%), increasing up to 3 °C and 5 °C for concentrations of 2.5% (wt%) of β-CD–thymol and β-CD–carvacrol, respectively, modifying the cold crystallization behavior of the PLA, and in consequence, the formation of the ordered structure of polymer matrix [29].

As can be seen in Figure 5, an endothermic peak was observed for all samples at a melting temperature, Tm, close to 168.5 °C, with slight temperature variations (lower than 1 °C), for PLA containing β-CD–carvacrol complexes. The little variations of cold crystallization and melting temperatures observed, when increasing β-CD–carvacrol and β-CD–thymol added to the PLA matrix, could be due to the increase in the chain mobility of the polymer matrix.

### 3.5. Thermogravimetry (TGA)

The thermal stability of PLA trays fortified with β-CD–carvacrol and β-CD–thymol complexes and non-fortified trays was measured by TGA, and all the samples had two weight loss steps (Figure 6).

A significant mass loss between 320 °C and 390 °C could be observed, which is in agreement with PLA decomposition, following which the thermal analysis curves slow down from 390 °C up to 500 °C till the complexes achieve a constant mass. In addition, pure PLA has a slightly higher stability (Figure 6) than PLA–β-CD–carvacrol and PLA–β-CD–thymol and the thermal stability of the polymeric matrices diminishes with increasing concentrations of β-CD–carvacrol or β-CD–thymol. These results indicate that although all polymer samples are, in essence, thermally stable below 300 °C, the mixtures containing β-CD–carvacrol and β-CD–thymol have a faster weight loss rate than pure PLA at the same temperature.

In practice, the PLA-based packages used in the food industry will be at room temperature or lower; therefore, the thermal stability of the PLA–β-CD–carvacrol and PLA–β-CD–thymol materials developed herein, will not be compromised.

### 3.6. Antifungal Assays

The antifungal properties of PLA packages containing monoterpene–cyclodextrin complexes was monitored for 10 days of incubation to determine their prospective potential uses in the agrofood industry, and the results are shown in Table 6.

As can be seen in Table 6, the results showed that PLA packages containing 2.5% and 5% β-CD–carvacrol or 5% β-CD–thymol (wt%) completely inhibited *Alternaria alternata* after 10 days of incubation (see Figure 7).

These results are in agreement with those obtained by Llana-Ruiz-Cabello et al. (2016), revealing the antimicrobial properties, against yeasts and moulds, of PLA films containing 5% and 10% of oregano essential oil in ready-to-eat salads [30].

The addition of carvacrol and thymol encapsulates to the polymeric matrix of PLA, as well as the optimized injection temperature (180–190 °C) to produce the packaging material, allow to play an active role against the growth of moulds. The inhibitory effect of these vapor phase assets can be attributed to the accumulation of volatile substances in the medium, followed by interaction with the hydrophobic portion of the cell membrane [31].

Other investigations associate the inhibitory effect of the active compounds of essential oils such as carvacrol and thymol with changes in the morphology of hyphae due to penetration of active compounds into the plasma membrane [32]. The antifungal activity of both the compounds was preserved following the inclusion and injection process, due to encapsulation with β-CD. Previous investigations with similar encapsulation processes such as spray drying [33], freeze drying [34], and lyophilization [35] have shown that encapsulation helps to preserve the antimicrobial and antioxidant properties, mainly of essential oils, and that they are advantageous because they improve water solubility by forming inclusion complexes.

Carvacrol and thymol are volatile compounds; therefore, they could be highly effective in removing bacteria from packaging [36]. The antimicrobial action of carvacrol and thymol released by the PLA matrix against a wide range of phytopathogens constitutes an interesting topic for further studies. Indeed, studies are being conducted regarding the measurement of the effectiveness of packaging against other microorganisms in different food products at various storage temperatures by this research group.

## 4. Conclusions

In this work, PLA packages filled with thymol or carvacrol complexed in β-cyclodextrins (β-CDs) were prepared and characterized to evaluate their potential use as antibacterial materials. The results obtained by FTIR confirm that the inclusion of carvacrol and thymol in the apolar cavity of β -CDs yielded a significantly higher efficiency of complexation for carvacrol (105.6) than for thymol (69.3). Different proportions of β-CD–thymol or β-CD–carvacrol (0.0%, 1.5%, 2.5%, and 5.0%, wt%) complexes were mixed with PLA for packaging development by injection process, selecting 180–190 °C as the optimal temperature. The presence of β-CD–carvacrol or β-CDs–thymol complexes confer to polymer material plasticizers features that diminish intermolecular forces of the polymer chains, thereby reducing packaging stiffness. In TGA experiments for thermal behavior analysis, the presence of thymol– or carvacrol–β-cyclodextrins solid complexes in PLA formulations slightly decreased the thermal degradation temperature of the polymer, when compared with pure PLA. The performance of the developed polymer materials against *Alternaria alternata* inhibition after 10 days of incubation provided evidence for their potential use in agrofood industry, since packages containing 2.5% and 5% β-CD–carvacrol, or 5% β-CD–thymol, completely inhibited fungal growth. Additional studies are required to evaluate the diffusion and release kinetics of carvacrol and thymol complexes in the PLA polymer matrix during food storage.

## Figures and Tables

**Figure 1 polymers-11-01720-f001:**
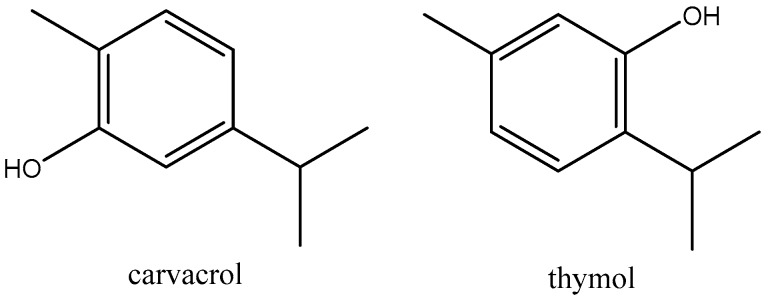
Chemical structures of carvacrol and thymol monoterpenes.

**Figure 2 polymers-11-01720-f002:**
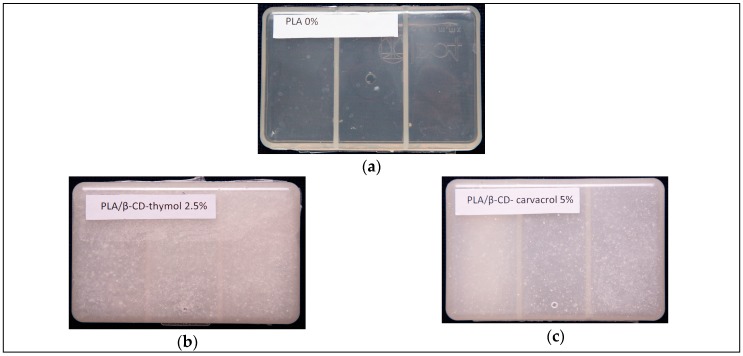
Boxes obtained by injection of the pellets. (**a**) (PLA); (**b**) (PLA/β-CD–thymol, 2.5%, wt%); (**c**) (PLA/β-CD–carvacrol, 5.0%, wt%).

**Figure 3 polymers-11-01720-f003:**
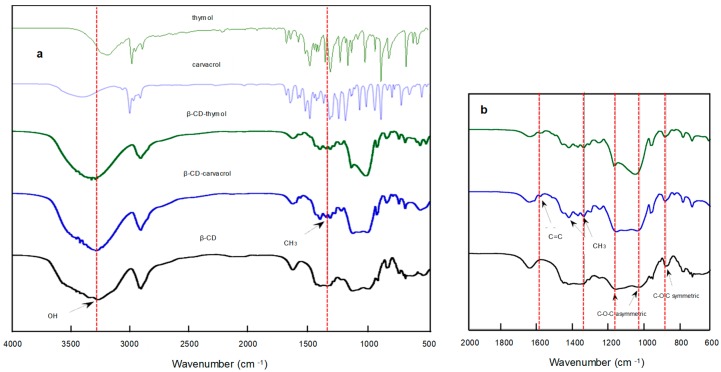
FTIR spectra of carvacrol (blue) and thymol (green); β-CD–thymol (green) and β-CD–carvacrol (blue) complexes, and β-CD (black) in normal (**a**) and broad view (**b**).

**Figure 4 polymers-11-01720-f004:**
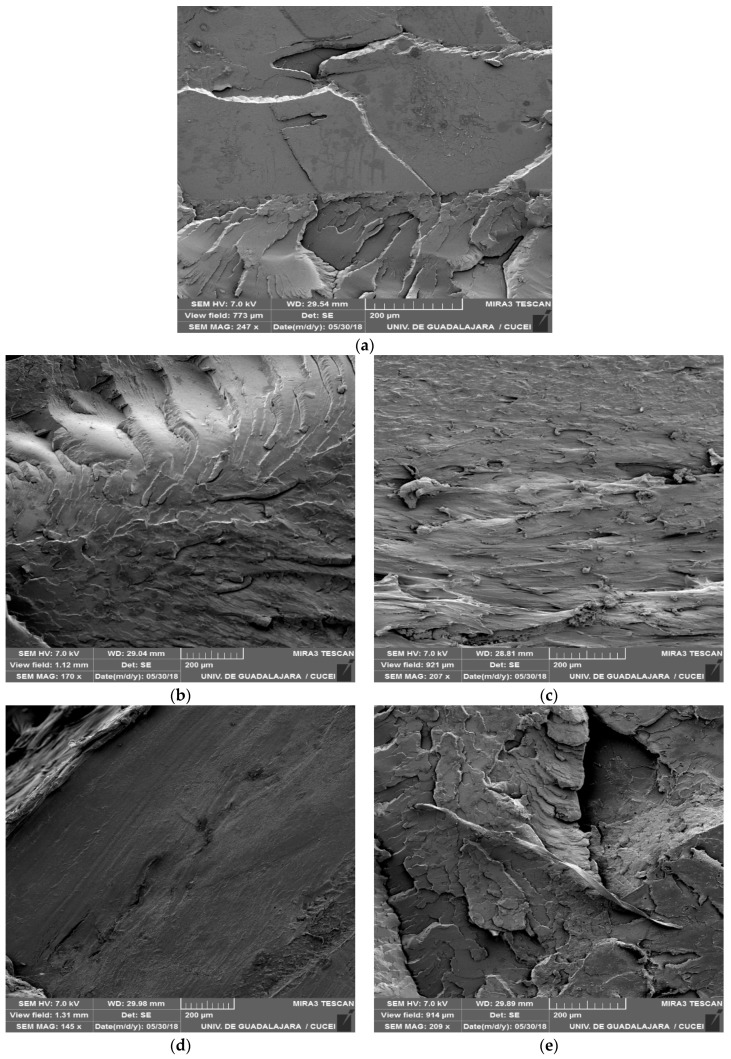
SEM micrographs of fracture samples: (**a**) 100% PLA; (**b**) 98.5% PLA with 1.5% β-CD–thymol; (**c**) 97.5% PLA with 2.5% β-CD–thymol; (**d**) 95.0% PLA with 5.0% β-CD–thymol; (**e**) 98.5% PLA with 1.5% β-CD–carvacrol; (**f**) 97.5% PLA with 2.5% β-CD–carvacrol; (**g**) 95.0% PLA with 5.0% β-CD–carvacrol.

**Figure 5 polymers-11-01720-f005:**
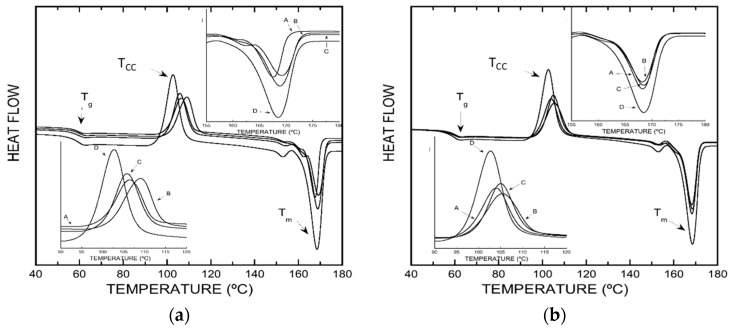
(**a**) DSC curves for: (**A**) PLA–(β-CD–carvacrol 1.5%, wt%); (**B**) PLA–(β-CD–carvacrol 2.5%, wt%); (**C**) PLA–(β-CD–carvacrol 5%, wt%); (**D**) PLA. (**b**) DSC curves for (**A**) PLA–(β-CD–thymol 1.5%, wt%); (**B**) PLA–(β-CD–thymol 2.5%, wt%); (**C**) PLA–(β-CD–thymol 5%, wt%); (**D**) PLA.

**Figure 6 polymers-11-01720-f006:**
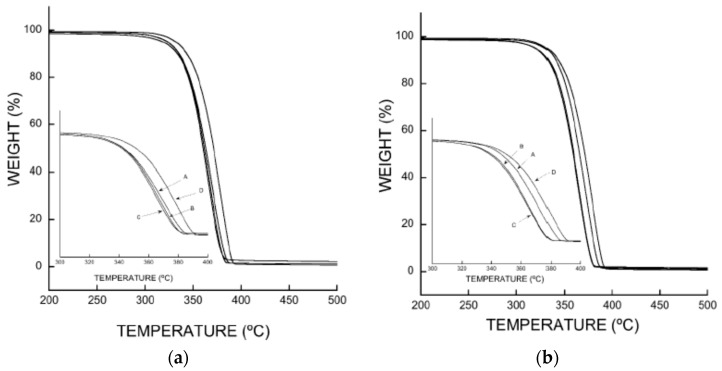
(**a**): Thermogravimetric analysis curves for (**A**) PLA–(β-CD–carvacrol 1.5%, wt%); (**B**) PLA–(β-CD–carvacrol 2.5%, wt%); (**C**) PLA–(β-CD–carvacrol 5, wt%); (**D**) PLA. (**b**): Thermogravimetric analysis curves for (**A**) PLA–(β-CD–thymol 1.5%, wt%); (**B**) PLA–(β-CD–thymol 2.5%, wt%); (**C**) PLA–(β-CD–thymol 5%, wt%); (**D**) PLA.

**Figure 7 polymers-11-01720-f007:**
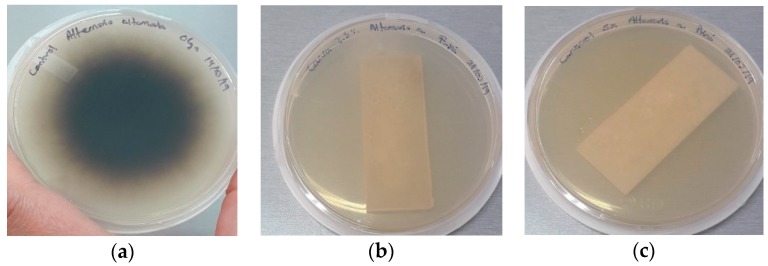
Photographs of the antifungal test: (**a**) Control *Alternaria alternata;* (**b**) PLA–β-CD–carvacrol packaging at 2.5%, wt%; (**c**) PLA–β-CD–carvacrol packaging at 5%, wt% with inoculation of *A. alternata* after 10 days of incubation.

**Table 1 polymers-11-01720-t001:** Carvacrol and thymol aqueous solubility (S_0_), complexation constant (Kc) with β-CDs, complexation efficiency (CE), and molar ratio (MR) at pH of 7.0.

Complexes	S_0_ (mmol L^−1^)	K_C_ (L mol^−1^)	CE (%)	Molar Ratio
Carvacrol/β-CDs	5.64 ± 0.12	1871 ± 143	105.6 ± 10.3	1:2
Thymol/β-CDs	5.77 ± 0.15 *	1198 ± 115	69.3 ± 9.2	1:2

* SD, standard deviation of triplicate determinations.

**Table 2 polymers-11-01720-t002:** Entrapment efficiency (EE) and process performance (PP) of β-CD–carvacrol and β-CD–thymol complexes in solid state.

Monoterpene	β-CD	EE (%)	PP (%)
**Carvacrol**	11 mM	45 ± 2.5 *	84 ± 3.2
**Thymol**	11 mM	47 ± 1.8	86 ± 3.7

* SD, standard deviation of triplicate determinations.

**Table 3 polymers-11-01720-t003:** Mechanical properties of the enriched or not PLA trays with β-CD–carvacrol complexes.

PLA Boxes with Different % of β-CD–Carvacrol				
Parameter	0%	1.50%	2.50%	5%
**Young’s modulus (Mpa)**	2873 ± 176	2327 ± 170 *	2259 ± 53 *	1960 ± 110 *
**Maximum stress (MPa)**	63.6 ± 4.5	49.9 ± 6.5 *	51.3 ± 4.9 *	47.5 ± 5.1 *
**Breaking point (%)**	2.4 ± 0.4	2.7 ± 0.3 *	2.9 ± 0.2 *	3.2 ± 0.4 *

Results expressed in mean ± standard deviation of ten determinations; symbol (*) in the same file indicates significant differences (*p* < 0.05) according to Tukey’s test.

**Table 4 polymers-11-01720-t004:** Mechanical properties of the enriched or not PLA trays with β-CD–thymol complexes.

PLA Boxes with Different % of β-CD–Thymol				
Parameter	0%	1.50%	2.50%	5%
**Young’s modulus (Mpa)**	2873 ± 176	2667 ± 161 *	2382 ± 69 *	2394 ± 118 *
**Maximum stress (MPa)**	63.6 ± 4.5	57.9 ± 6.8 *	53.2 ± 2.3 *	55.1 ± 5.2 *
**Breaking point (%)**	2.4 ± 0.4	2.8 ± 0.3 *	2.9 ± 0.2 *	3.1 ± 0.3 *

Results expressed in (mean ± standard deviation) of ten determinations; Symbol (*) in the same file indicate significant differences (*p* < 0.05) according to Tukey’s test.

**Table 5 polymers-11-01720-t005:** Parametric values of DSC obtained from pure PLA and added β-CD–carvacrol or β-CD–thymol at 1.5%, 2.5%, and 5%, wt%.

Parameter	Control * 0%	PLA–Thymol–β-CDs (wt%)			PLA–Carvacrol–β-CDs (wt%)		
		1.5%	2.5%	5.0%	1.5%	2.5%	5.0%
**Tg (°C)**	59	61	59	61	60	60	60
**Tcc (°C)**	102.7	103.8	105.4	105.0	106.5	107.7	105.9
**Tm (°C)**	168.5	168.5	168.5	168.8	167.9	168.8	169.1
**ΔHc Energy (J/g)**	36.09	33.08	30.61	29.03	36.42	32.09	36.83
**ΔHm Energy (J/g)**	45.63	44.09	37.44	35.65	44.72	37.54	42.95

* Control, pure PLA without β-CD–carvacrol or β-CD–thymol; Tg, glass transition temperature; Tcc, cold crystallization transition temperature; Tm, melting transition temperature.

**Table 6 polymers-11-01720-t006:** Antifungal activity over incubation time of developed packaging materials against *Alternaria alternate*.

Type of Packaging	Encapsulation Concentration (% *w*/*w*)	Incubation Time	
		5 Days	10 Days
**PLA-control**	0.0%	29.7 ^a^	69.0
**PLA–β-CD–thymol**	1.5%	28.3	71.6
	2.5%	30.0	60.0
	5.0%	3.3 *	0.0 *
**PLA–β-CD–carvacrol**	1.5%	29.7	65.3
	2.5%	0.0 *	0.0 *
	5.0%	0.0 *	0.0 *

^a^ Diameter of the colony or surface area in mm. For each test, * values are statistically significant (*p* < 0.05).

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
