# Peer review of "Development and Characterization of a Biodegradable PLA Food Packaging Hold Monoterpene–Cyclodextrin Complexes against Alternaria alternata"

_polymers, 2019, doi:10.3390/polym11101720_

Round 1

Reviewer 1 Report

I had a great pleasure to read and review the manuscript. I think the manuscript is appropriate for publication in the Polymers, and can be accepted as it is. 

Author Response

Thank you for your constructive comments

Reviewer 2 Report

In this work, the authors developed PLA based food packaging materials containing β-CD-carvacrol, β- CD-thymol with anti-Alternaria alternate properties.  there is a high demand for biobased food packing materials and this study evaluated a bio based packaging which is also conferred with some antimicrobial properties which make it very interesting. That being said this article adds useful information to the body of the literature. However, there some issues that need to be addressed before publication. One major point is the lack of clarity in the message due to poor English. The authors need to carefully revise the manuscript to better deliver the message. Here are some of my comments.

Examples of English issue:

Line 17-18:

The most abundant and important WHAT?= bacteria

The first 3 lines of the abstract need to be revised and rewritten to improve the clarity.

Line 160:  that were introduced in a pilot extruder by us developed,

Inline 219: please identify the groups/data sets that were subjected to statistical analysis

The Provided FTIR peaks are very hard to identify and interpret, I suggest to magnify those sections in a separate image and provide a clear control to better read and understand the differences in the chemical compositions of the samples.

Table 3. was there a significant difference in the young modulus values of PLA boxes with 0% and 5% B-CD? Currently, it looks like the differences only exist between 2.5 and 5%. Also for the maximal stress values e.g. between 0% and 5%?

Comparing the values of table 3 and 4, What is the rationale behind the different mechanical values observed for complexes containing B-CD-carvacrol or b-CD-thymol?

Figure 3. the scale bar is not an easy read. Please modify.

Author Response

Reviewer 2# comments:

In this work, the authors developed PLA based food packaging materials containing β-CD-carvacrol, β- CD-thymol with anti-Alternaria alternate properties. There is a high demand for biobased food packing materials and this study evaluated a bio based packaging which is also conferred with some antimicrobial properties which make it very interesting. That being said this article adds useful information to the body of the literature. However, there some issues that need to be addressed before publication.

Thank you for your constructive comments. Next, we answer to the asked questions:

One major point is the lack of clarity in the message due to poor English. The authors need to carefully revise the manuscript to better deliver the message. Here are some of my comments:

Examples of English issue:

(1). Line 17-18: The most abundant and important WHAT?= bacteria

As suggested by reviewer, fungi have been included the abstract to clarify readers.

(2). The first 3 lines of the abstract need to be revised and rewritten to improve the clarity.

The abstract part has been revised and modified based on the recommendations made by the reviewer and highlighted in red.

(3). Line 160:  that were introduced in a pilot extruder by us developed,

Sentence “by us developed” was deleted from the text

(4). Inline 219: please identify the groups/data sets that were subjected to statistical analysis

As suggested by reviewer, data subjected to statistical analysis have been identified in section 2.8.

(5). The Provided FTIR peaks are very hard to identify and interpret, I suggest to magnify those sections in a separate image and provide a clear control to better read and understand the differences in the chemical compositions of the samples.

As suggested by reviewer, FTIR of carvacrol and thymol has been magnified and included in Figure A, to better understand the differences between free and complexed monoterpenes.

(6). Table 3. was there a significant difference in the young modulus values of PLA boxes with 0% and 5% B-CD? Currently, it looks like the differences only exist between 2.5 and 5%. Also for the maximal stress values e.g. between 0% and 5%?

As suggested by reviewer, symbol (*) has been included in Table 3 (and Table 4), to better understanding mechanical properties significant differences regarding the control (0%).

(7). Comparing the values of table 3 and 4, What is the rationale behind the different mechanical values observed for complexes containing B-CD-carvacrol or b-CD-thymol?

The CE values obtained for carvacrol-?-CDs was 105.6%, approximately 65% higher than the value obtained for thymol-?-CDs 69.3%. This high value, above 100%, indicates that at pH 7.0, there are more ?-CDs complexing carvacrol than free in solution, being lower the number of ?-CDs in the case of thymol, since CE is lower than 100%. Considering that mechanical properties of the enriched PLA gradually decreases as the concentration (weight percentage, wt%), of the dehydrated complexes increases, obtaining for example the lowest value of Young's modulus in the sample fortified with 5% of carvacrol (1960); 2394 for thymol (in the same experimental conditions); this fact could be due to the higher CE value obtained for carvacrol-?-CDs.

A paragraph has been included in the manuscript explaining this fact

(8). Figure 3. the scale bar is not an easy read. Please modify.

As suggested by reviewer, SEM image has been magnify for a better appreciation of the scale bar.

Reviewer 3 Report

The effects of entrapped monoterpenes on the thermal and mechanical properties and antifungal activity were investigated. If the following minor issues are improved and the novelty of the paper is intensified, the paper can be published in Polymers.

(1) Chemical structures of two monoterpenes should be shown for the researchers in the polymer science and technology.

(2) The explanation why the genus Alternaria is used for antifungal activity should be stated.

(3) The averaged molecular weight (Mor Mn), poly dispersity index (Mw/Mn), and D-or L-lactate unit of PLA should be given.

(4) "cold crystallization temperature (Tcc)"rather than "crystallization temperature (Tc)" should be used in the manuscript.

(5) In Figure 2 panel B, for the uniformity of expressing the sample names, "ß-CD" should be used for "ß-Cyclodextrin".

(6) In Table 5, for the uniformity of expressing the concentrations of monoterpenes "PLA 100%" should be rephrased considering the expression in Tables 3 and 4.

(7) "Poly lactic acid"should be expressed as "Poly(lactic acid)".

(8) Considering the Tc results in Table 5, A, B, C and D in Figure 4 do not correspond to (a), (b)< (c) and (d) in the figure caption of Table 4.

Author Response

Reviewer 3# comments:

The effects of entrapped monoterpenes on the thermal and mechanical properties and antifungal activity were investigated. If the following minor issues are improved and the novelty of the paper is intensified, the paper can be published in Polymers.

Thank you for your constructive comments. Next, we answer to the asked questions:

(1) Chemical structures of two monoterpenes should be shown for the researchers in the polymer science and technology.

As suggested by reviewer, carvacrol and thymol chemical structures has been included in materials and method section (Figure 1), renamed properly the rest of figures.

(2) The explanation why the genus Alternaria is used for antifungal activity should be stated.

As suggested by reviewer, a brief explanation of the widely food and feed contamination by mycotoxins produced by Alternaria species, justifying their selection as model microorganism, has been included in introduction section. Also, a new reference has been included, renamed properly all references.

França, K.R.S.; Silva, T.L.; Cardoso, T.A.L.; Ugulino, A.L.N.; Rodrigues, A.P.M.; De Mendonça Júnior, A.F. In vitro Effect of Essential Oil of Peppermint (Mentha x piperita L.) on the Mycelial Growth of Alternaria alternata. J. Exp. Agric. 2018, 26(5), 1-7. https://doi.org/10.9734/JEAI/2018/44243.

(3) The averaged molecular weight (Mw or Mn), poly dispersity index (Mw/Mn), and D-or L-lactate unit of PLA should be given.

As suggested by reviewer, averaged molecular weight (Mw), poly dispersity index and D-L isomer content of PLA has been included in materials and method section.

(4) "cold crystallization temperature (Tcc)"rather than "crystallization temperature (Tc)" should be used in the manuscript.

As suggested by reviewer, "crystallization temperature (Tc)" has been changed by "cold crystallization temperature (Tcc)" in all the manuscript and Figure 4.

(5) In Figure 2 panel B, for the uniformity of expressing the sample names, "ß-CD" should be used for "ß-Cyclodextrin".

As suggested by reviewer, ß-Cyclodextrin has been changed by ß-CD.

(6) In Table 5, for the uniformity of expressing the concentrations of monoterpenes "PLA 100%" should be rephrased considering the expression in Tables 3 and 4.

As suggested by reviewer (table 5), PLA 100% has been rewritten, in line with the terminology used in tables 3 and 4. In addition, a footnote has been included to clarify readers.

(7) "Poly lactic acid" should be expressed as "Poly(lactic acid)".

As suggested by reviewer, "Poly lactic acid" has been changed by "Poly(lactic acid)" in all the manuscript

(8) Considering the Tc results in Table 5, A, B, C and D in Figure 4 do not correspond to (a), (b)< (c) and (d) in the figure caption of Table 4.

I apologize, but by mistake in Figure 4 the same graph corresponding to the DSC of carvacrol was included. It has already been remedied, and DSC of thymol was included on the right.